# COVID-19 gender susceptibility and outcomes: A systematic review

Ines Lakbar [1,2,3], David Luque-Paz[4,5], Jean-Louis Mege[2], Sharon Einav[6], Marc Leone[1,2]*

1 Department of Anesthesiology and Intensive Care Unit, Nord Hospital, Assistance Publique Hôpitaux Universitaire de Marseille, Aix Marseille University, Marseille, France, 2 MEPHI, IHU Méditerranée Infection, Aix Marseille Université, Marseille, France, 3 Department of Anesthesiology and Intensive Care Unit, Toulouse, France, 4 Infectious Diseases and Intensive Care Unit, Pontchaillou University Hospital, Rennes, France, 5 University of Rennes, Inserm, BRM (Bacterial Regulatory RNAs and Medicine), UMR 1230, Rennes, France, 6 Intensive Care Unit of the Shaare Zedek Medical Centre and Hebrew University Faculty of Medicine, Jerusalem, Israel

☯ These authors contributed equally to this work.
* marc.leone@ap-hm.fr

## Abstract

### Background

Epidemiological differences between men and women have been reported with regards to sepsis, influenza and severe coronavirus infections including SARS-CoV and MERS-CoV.

### Aim

To systematically review the literature relating to men versus women on SARS-CoV-2 in order to seek differences in disease characteristics (e.g. infectivity, severity) and outcomes (e.g. mortality).

### Methods

We searched 3 electronic databases up or observational studies reporting differences between men and women in the SARS-CoV-2 disease characteristics stated. We identified and included 47 studies, reporting data for 21,454 patients mainly from China.

### Results

The unadjusted mortality rates of men were higher than those of women, with a mortality OR 0.51 [0.42, 0.61] (p<0.001) for women. The proportion of men presenting with severe disease and admitted to the intensive care unit (ICU) was also higher than that of women (OR 0.75 [0.60–0.93] p<0.001 and OR 0.45 [0.40–0.52] p<0.001 respectively). Adjusted analyses could not be conducted due to lack of data.

### Conclusion

COVID-19 may be associated with worse outcomes in males than in females. However, until more detailed data are provided in further studies enabling adjusted analysis, this remains an unproven assumption.

**Data Availability Statement:** All relevant data are within the manuscript and its Supporting Information files.

**Funding:** This work was supported by the department of anesthesiology and intensive care unit of Nord Hospital, Marseille, France.

**Competing interests:** IL, DLP, SE and JLM have no conflict of interest to disclose. ML served as speaker for MSD, Pfizer and as consultant for Amomed, Aguettant and Gilead. This does not alter our adherence to PLOS ONE policies on sharing data and materials.

# Introduction

Many infectious diseases, including sepsis, have been associated with gender differences in disease incidence, morbidity and mortality [1–3]. Epidemiological differences between men and women have also been reported with regards to previous outbreaks of highly-pathogenic coronaviruses such as the severe acute respiratory syndrome coronavirus (SARS-CoV-1) [4] and the Middle East respiratory syndrome coronavirus (MERS-CoV) [5]. Men are more likely to be infected with MERS-CoV and SARS-CoV than women and when infected, they also seem to have worse outcomes [4]. These clinical findings are consistent with mouse models suggesting that estrogen reduces the susceptibility and severity of SARS-CoV infection; female mice that had their ovaries removed suffered from increased disease infectivity and severity [6]. SARS-CoV-2 may also affect men and women differently. However, at the time of this writing few epidemiological studies have explored the data on SARS-CoV-2 with regards to COVID-19 disease incidence and case-fatality rate. It therefore remains unclear if the disease characteristics (e.g. infectivity, severity) and outcomes (e.g. mortality) of patients with SARS-CoV-2 infection differ between the genders [7].

The effects of different viral infections on men and women are probably pathogen–specific. For instance, contrary to reports on MERS-CoV and SARS-CoV-1, women have been reported to suffer higher morbidity and mortality than men during influenza outbreaks [8]. Biological factors (referred to as sex-related variables) and sociocultural factors (referred as gender-related variables) are often put forward as explanations for the differences observed between men and women with regards to susceptibility and host response [9–12]. In this review we aimed to compile the data relating to men versus women in the literature on SARS-CoV-2. The intention was to seek differences, if these exist, between men and women in SARS-CoV-2 disease characteristics (e.g. infectivity, severity) and outcomes (e.g. mortality). Despite the nuances in terminology described above, the term "gender" is used throughout the manuscript in relation to all variables for the sake of convenience.

# Methods

This study was conducted in accordance with the Preferred Reporting Items for Systematic Reviews and Meta-Analyses (PRISMA) recommendations [13] and was registered in the PROSPERO database prior to study initiation (CRD42020184142).

## PICO question

We sought to study whether among adult patients with COVID-19 (P) women ("I") differ from men (C) with regards to disease characteristics (e.g. infectivity, severity) and outcomes (any outcomemortality, severity and ICU admission rates) (O).

## Search strategy

Two of the authors (IL, DLP) conducted a systematic search of PubMed, Medline, Web of Science and the Cochrane Library databases from inception to 1-June-2020 for studies describing any epidemiological characteristic related to patient gender in COVID-19. We initially used a broad search strategy which included any paper regarding the disease at hand. The following search terms were used: "2019 novel coronavirus" OR "SARS-CoV-2" OR "2019-nCoV" OR "novel coronavirus". We then restricted the search for articles on adult humans. In the last stage of the database search, articles which were not written in English and those which were conducted on animal models and special populations (children, pregnant women) were also

excluded. Among the articles remaining we sought those providing any information regarding gender.

## Eligibility criteria and study selection

We included randomized controlled trials, clinical trials, observational cohorts and case series that (1) described adult patients with SARS-CoV-2 infection confirmed by real-time reverse transcriptase polymerase chain reaction (rRT-PCR) and also (2) provided information regarding the relative proportion of men and women with confirmed disease, and/or admitted to hospital and/or the intensive care unit (ICU) and as well as their mortality rates. We also sought information regarding the background and acute disease characteristics of the two populations (men and women) with particular emphasis on severity of disease. We did not include preprints.

The articles were first selected by two of the authors (IL, DLP) in duplicate and independently based on title and abstract. Those selected were downloaded in full to allow full text review and those fulfilling the predetermined inclusion criteria regarding both disease diagnosis and presentation of relevant epidemiological data were included. The reference lists of relevant articles were also screened (i.e. snowballing method). Disagreement over study inclusion was resolved by consensus or, if necessary, by the adjudication of a third author (ML). For the sake of comprehensiveness, we decided to include also relevant single-arm studies without a comparison group. We excluded articles reporting data with complete or partial overlap with other reports. Abstracts, conference proceedings, and publications describing a single treatment arm were included only if they presented sufficient details. The exact details of the inclusion/exclusion process are shown in the PRISMA diagram (Fig 1).

As we also intended to perform an individual patient level meta-analysis (IPDMA), we contacted the corresponding authors of all the included studies (and five of the excluded studies to ensure full data capture) through electronic mail. The authors were requested to share their anonymised data regarding comorbidities, treatment with antivirals, treatment with corticosteroids and age of deceased among men and women. IPDMA allows for a more exact analysis of adjusted outcome. We intended to study in particular the adjusted primary outcome of mortality at any time.

## Assessment of risk of bias (RoB)

Two of the authors (IL, DLP) assessed the RoB of the included studies independently and in duplicate. Disagreements over RoB were resolved by consensus or, if necessary, adjudicated by a third author (ML). As in all of the included studies the data were observational and the groups were not randomly assigned we used the Newcastle Ottawa Scale for assessing RoB and did not evaluate the "comparability" item [14]. For each domain we rated the overall RoB as the highest risk attributed to any criterion. A good quality score required at least two stars in the selection domain and three stars in the outcomes domain. The quality score was considered poor if the study obtained zero or one star in the selection and outcomes domains. A poor quality score was determined as no or one star in selection and outcomes domains. Since the element of comparability could not be assessed, the absence of a star in this element was not decisive for a good quality rating for a study. We assessed certainty in overall effect estimates using GRADE (Grade of Recommendations Assessment, Development and Evaluation) methods [15].

## Data extraction

Two of the authors (IL, DLP) extracted data from the included studies independently and in duplicate to pre-prepared forms. The data extracted included the characteristics of the

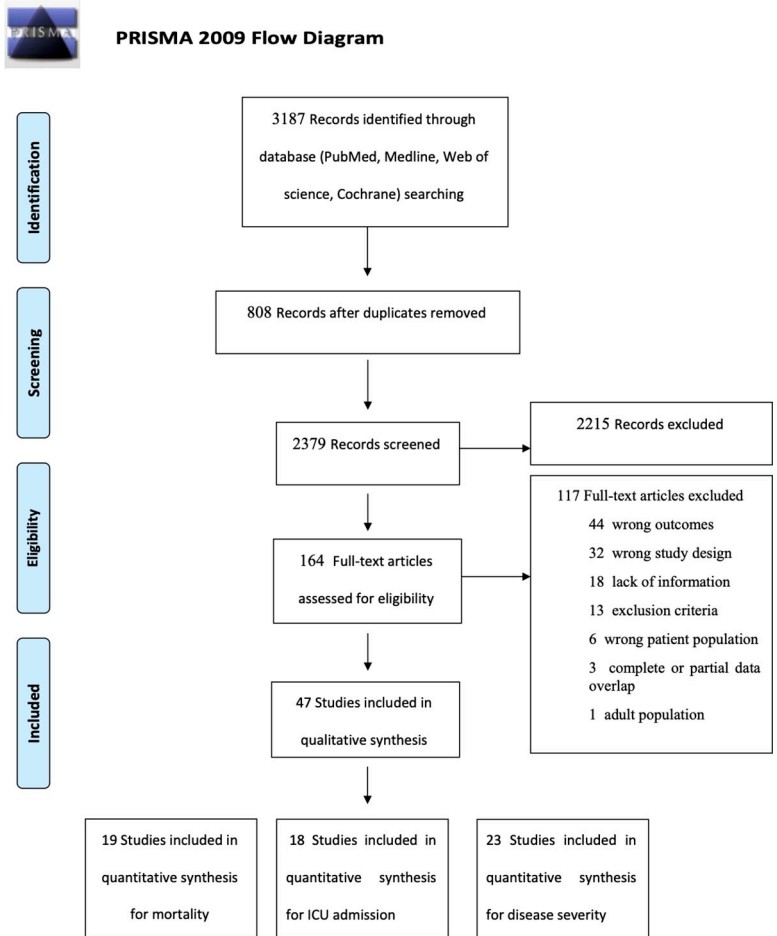

**Fig 1. Study selection process.**

included publication (e.g. type of publication, journal name, date of publication) and the information provided (e.g. patients' age, number of men and women reported infected, ill [hospital admissions] and critically ill [ICU admissions], patients' background characteristics, disease severity and mortality).

## Outcome measures

As we were unsure whether the authors approached would be forthcoming with their data we set our main outcome measure as the unadjusted all-cause mortality rate, as reported in the included studies (hospital, out-of-hospital, depending on the duration of the follow-up). Secondary outcomes included disease severity (as per author definition), and ICU admission rates as well as the adjusted mortality rates.

## Statistical analysis

We set out to perform a quantitative synthesis if two or more studies were identified with sufficient homogeneity in study design, interventions and outcomes and low risk of bias [16]. The odds ratios (OR) were calculated by reporting available data from included studies, with 95% confidence intervals (95% CI). P-values were considered significant if <0.05. Pooled

prevalences and their 95% CIs were used to summarize the weighted effect size for each study grouping variables using the binary fixed-effects model (Mantel-Haenszel method). Statistical heterogeneity was evaluated by the $I^2$, corresponding to the proportion of total variation due to inter-study heterogeneity, and visual inspections of the forest plots. All the analyses were performed using Revman Version 5.4 (Copenhagen, Denmark).

## Results

A total of 2,379 studies were screened, of which 47 contributed data to our analysis for a total of 21,454 patients consisting of 11,176 women and 10,278 men [17–61]. Most of the included studies originated from different hospitals in China (n = 38), one originated from Hong Kong, two from South Korea, five from Europe and North America and one from Israel (Table 1). The quality of 43 of the studies was rated as poor and of 4 of the studies was rated as good (Table 2).

### Crude mortality rates

**Characteristics of the studies that contributed data to the analysis of crude mortality rates in men and women.**   The crude mortality rates were reported separately for 5,589 men and 6,751 women in 19 studies. Among the included studies, 1 was prospective and observational, 10 were retrospective database studies, and 9 were case series. None of the studies aimed a-priori to compare between the genders as the main outcome measure. There were no studies that attempted to assess sex differences as the main outcome measure. Most of the studies included less than 300 patients. The studies reporting crude mortality data involved mainly cohorts of hospitalized patients, except for one Korean epidemiological study of 7,755 patients [61]. Variables such as age, patient comorbidities, and treatments were not reported individually, so mortality data could not be adjusted. The characteristics and the main findings of the included studies are provided in Table 1.

**RoB in the studies that contributed data to the analysis of mortality rates in men and women.**   The results of the quality assessment were highly variable ranging from 1/7 (two studies) to 5/7 (two studies) on the Newcastle Ottawa Scale (Table 2). The main caveats were that patient follow-up was either non-existent or very short, and mortality assessment was not standardized.

**Mortality rates in men and women.**   Overall 280 (4%) women died versus 623 (11%) men (p<0.001, OR 0.51 [0.42, 0.61] for women versus men, see Fig 2) based on a very low-certainty evidence (Table 3). However, there was significant variability in the mortality rates reported in different studies, with OR and 95% CI ranging from 0.14 [0.01, 2.86] to 1.33 [0.27, 6.50]. Heterogeneity reported as $I^2$ was low at 6% for this analysis. The funnel plot (S1a Fig in S1 File) and the Egger test (p = 0.54) both indicated that there was no evidence for publication bias.

**Sensitivity analyses.**   A sensitivity analysis was performed to assess the stability of the results. Heterogeneity as measured by $I^2$ was already low, therefore omitting studies sequentially had no significative influence on the results (S2 Fig in S1 File).

### Unadjusted severe disease presentation

**Characteristics of the studies that contributed data to the analysis of severe disease presentation in men and women.**   The proportion of men and women with severe disease presentation was reported in 23 studies, including 1,721 women and 1,970 men. Among the included studies 1 was prospective observational, 9 were retrospective database studies, and 13 were case series. None of the studies aimed a-priori to compare between the genders as the severity of the disease. Most of the studies included less than 200 patients. As for the definition

**Table 1. Studies included in the systematic review.**

| Newcastle Ottawa Scale (Total grade) | Author | Journal | Centre (country) | Design | | | No. of patients | | | Setting | Age (+/-SD or IQR) | Crude inhospital mortality | | ICU admission | | Severity | | Severity definition |
|---|---|---|---|---|---|---|---|---|---|---|---|---|---|---|---|---|---|---|
| | | | | | | | Total | F | M | | | F | M | F | M | F | M | |
| 3/7 | Chen G | J Clin Invest. | 1 (China) | Retrospective | Case series | | 21 | 4 | 17 | Hospital | 56 (50–65) | 0 | 4 | - | - | 1 | 10 | National Health Commission of China |
| 5/7 | Chen R | Chest | 575 (China) | Retrospective | Cohort | | 1590 | 725 | 865 | Hospital | NA | 11 | 39 | - | - | - | - | NA |
| 4/7 | Chen T | BMJ | 1 (China) | Retrospective | Case series | | 274 | 103 | 171 | Hospital | 62 (44–70) | 30 | 83 | - | - | - | - | National Health Commission of China |
| 3/7 | Chen X | Clin Infect Dis. | 1 (China) | Retrospective | Case series | | 48 | 11 | 37 | Hospital | 64.6 (+/-18.1) | - | - | 2 | 15 | 1 | 9 | National Health Commission of China |
| 3/7 | Conversano A | Hypertension | 1 (Italy) | Retrospective | Consecutive case series | | 191 | 60 | 131 | Hospital | 63.4 (+/-14.9) | 11 | 31 | - | - | - | - | NA |
| 3/7 | Gautret P | Travel Med Infect Dis | 1 (France) | Retrospective | Cohort | | 80 | 37 | 43 | Hospital | 52 (18–88) | 0 | 1 | 0 | 3 | - | - | NA |
| 4/7 | Guan W | N Engl J Med | 552 (China) | Retrospective | Cohort | | 1099 | 459 | 640 | Hospital | 47 (35–58) | - | - | 22 | 45 | 73 | 100 | American Thoracic Society Guidelines |
| 3/7 | Hong KS | Yonsei Med J | 1 (South Korea) | Retrospective | Cohort | | 98 | 60 | 38 | Hospital | 55.4 +/-17.1 | - | - | 7 | 6 | - | - | Critically ill based on ARDS definition |
| 3/7 | Hou H | Clin Exp Immunol | 1 (China) | Retrospective | Cohort | | 389 | 189 | 200 | Hospital | 61.3 (+/-13.8) | - | - | 20 | 32 | 75 | 94 | National Health Commission of China |
| 3/7 | Huang C | Lancet | 1 (China) | Prospective | Observational cohort study | | 41 | 11 | 30 | Hospital | 49 (41–58) | - | - | 2 | 11 | - | - | WHO interim guidelines |
| 3/7 | Itelman E | Isr Med Assoc J | 1 (Israel) | Retrospective | Cohort | | 162 | 57 | 105 | Hospital | 52 (+/-20) | - | - | - | - | 5 | 21 | Clinical definition |
| 1/7 | Korea—CDC | Osong Public Health Res Perspect | multicentre (Korea) | Retrospective | Case series | | 7755 | 4808 | 2947 | Epidemiological | NA | 29 | 37 | - | - | - | - | NA |
| 2/7 | Lagi F | Euro Surveill | 1 (Italy) | Retrospective | Cohort | | 84 | 29 | 55 | Hospital | 62 (51–72) | - | - | 2 | 14 | - | - | Critically ill based on ARDS definition |
| 5/7 | Liu F | Theranostics | 1 (China) | Retrospective | Cohort | | 134 | 71 | 63 | Hospital | 51.5 (37–65) | - | - | - | - | 4 | 15 | National Health Commission of China |
| 4/7 | Liu F | J Clin Virol | 1 (China) | Retrospective | Cohort | | 140 | 91 | 49 | Hospital | 65.5 (54.3–73) | - | - | - | - | 25 | 8 | National Health Commission of China |
| 4/7 | Liu J | EBioMedicine | 1 (China) | Retrospective | Cohort | | 40 | 25 | 15 | Hospital | 48.7 (+/-13.9) | - | - | - | - | 6 | 7 | National Health Commission of China |
| 3/7 | Liu K-C | Eur J Radiol | 6 (China) | Retrospective | Case series | | 73 | 32 | 41 | Hospital | 41.6 (+/-14.5) | - | - | 3 | 0 | 11 | 10 | National Health Commission of China |
| 3/7 | Liu Z | Korean J Radiol | 3 (China) | Retrospective | Case series | | 72 | 33 | 39 | Hospital | 46.2 (+/-15.9) | 0 | 0 | - | - | 5 | 3 | WHO interim guidelines |
| 1/7 | Marullo AG | Minerva Cardioangiol | 90 (Europe) | Retrospective | Case series | | 217 | 46 | 171 | Hospital | 52 (+/-11) | 6 | 26 | 46 | 171 | - | - | NA |
| 3/7 | Pan L | Am J Gastroenterol | 3 (China) | Retrospective | Case series | | 103 | 48 | 55 | Hospital | 52.21 (+/-15.92) | - | - | 7 | 16 | 7 | 7 | NA |
| 2/7 | Petrilli C | BMJ | 1 (USA) | Prospective | Observational cohort study | | 5279 | 2664 | 2615 | Hosp/ER | 54 (38–66) | - | - | 334 | 656 | - | - | Critically ill based on ARDS definition or death |
| 2/7 | Shi H | Lancet Infect Dis | 2 (China) | Retrospective | Case series | | 81 | 39 | 42 | Hospital | 49.5 (+/-11) | 0 | 3 | 1 | 10 | 4 | 6 | NA |
| 2/7 | Sun B | Emerg Microbes Infect | 1 (China) | Retrospective | Case series | | 38 | 14 | 24 | Hospital | 44 (32–56) | - | - | - | - | - | - | NA |
| 3/7 | Sun H | J Am Geriatr Soc | 1 (China) | Retrospective | Case-control study | | 244 | 111 | 133 | Hospital | 69 (65–76) | 39 | 82 | - | - | - | - | NA |
| 3/7 | Sun S | Clin Chim Acta | 1 (China) | Retrospective | Case series | | 116 | 56 | 60 | Hospital | 50 (41–57) | - | - | 3 | 6 | 9 | 18 | National Health Commission of China |
| 4/7 | Tang N | J Thromb Haemost | 1 (China) | Retrospective | Cohort | | 449 | 181 | 268 | Hospital | 65.1 (+/-12) | 44 | 90 | - | - | - | - | National Health Commission of China |
| 3/7 | Tian S | J Infect | 57 (China) | Retrospective | Cohort | | 262 | 135 | 127 | Hospital | 47.5 | 1 | 2 | - | - | 20 | 26 | Clinical definition |
| 5/7 | To K | Lancet Infect Dis | 2 (Hong Kong) | Prospective | Observational cohort study | | 23 | 10 | 13 | Hospital | 62 (37–75) | - | - | - | - | 4 | 6 | Clinical definition |
| 5/7 | Wan S | Br J Haematol | 1 (China) | Retrospective | Case series | | 123 | 57 | 66 | Hospital | 43 (+/-13.1) | - | - | 4 | 10 | 10 | 11 | National Health Commission of China |
| 5/7 | Wang D | Crit Care | 2 (China) | Retrospective | Case series | | 107 | 50 | 57 | Hospital | 51 (36–65) | 3 | 16 | - | - | - | - | Clinical definition |
| 2/7 | Wang F | Endocr Pract | 1 (China) | Retrospective | Case series | | 28 | 7 | 21 | Hospital | 68.6 (+/-9) | - | - | - | - | - | - | National Health Commission of China |
| 3/7 | Wang J | Clin Radiol | 3 (China) | Retrospective | Case series | | 93 | 36 | 57 | Hospital | 52.1 (+/-18.1) | - | - | 6 | 14 | - | - | National Health Commission of China |
| 3/7 | Wang L | J Infect | 1 (China) | Retrospective | Consecutive case series | | 339 | 173 | 166 | Hospital | 69 (65–76) | 26 | 39 | - | - | - | - | National Health Commission of China |
| 3/7 | Wang L | Am J Nephrol | 1 (China) | Retrospective | Case series | | 116 | 49 | 67 | Hospital | 54 (38–69) | 0 | 2 | 5 | 6 | 19 | 27 | WHO interim guidelines |
| 3/7 | Wang M | Aging (Albany, NY) | 3 (China) | Retrospective | Cohort | | 66 | 23 | 43 | Hospital | NA | - | - | - | - | - | - | NA |
| 4/7 | Wu C | JAMA Intern Med | 1 (China) | Retrospective | Cohort | | 201 | 73 | 128 | Hospital | 51 (43–60) | 15 | 29 | 24 | 60 | - | - | Critically ill based on ARDS definition |
| 3/7 | Xu Y-H | J Infect | 1 (China) | Retrospective | Case series | | 50 | 21 | 29 | Hospital | 43.9 (+/-16.8) | - | - | 0 | 3 | 3 | 7 | National Health Commission of China |
| 4/7 | Yan Y | BMJ Open Diabetes Res Care | 1 (China) | Retrospective | Cohort | | 193 | 79 | 114 | Hospital | 69 (49–73) | 32 | 76 | - | - | 79 | 114 | National Health Commission of China |
| 3/7 | Yang A-P | Int Immunopharmacol | 1 (China) | Retrospective | Case series | | 93 | 37 | 56 | Hospital | 46.4 (+/-17.6) | - | - | - | - | 6 | 18 | American Thoracic Society Guidelines |
| 4/7 | Yang X | Lancet Respir Med | 1 (China) | Retrospective | Cohort | | 52 | 17 | 35 | Hospital | 59.7 (+/- 13.3) | 11 | 21 | 17 | 35 | - | - | Critically ill based on required mechanical ventilation |
| 3/7 | Yang Y | J Allergy Clin Immunol | 1 (China) | Retrospective | Case-control series | | 50 | 21 | 29 | Hospital | 62 (22–78) | - | - | 3 | 8 | 14 | 14 | National Health Commission of China |
| 4/7 | Yuan M | PloS One | 1 (China) | Retrospective | Consecutive case series | | 27 | 15 | 12 | Hospital | 60 (47–69) | 6 | 4 | - | - | - | - | NA |
| 2/7 | Zeng F | J Med Virol | 1 (China) | Retrospective | Case series | | 331 | 204 | 127 | Hospital | NA | - | - | - | - | 11 | 11 | NA |
| 3/7 | Zhang G | Respir Res | 1 (China) | Retrospective | Cohort | | 95 | 42 | 53 | Hospital | NA | - | - | 9 | 16 | 11 | 21 | National Health Commission of China |
| 3/7 | Zheng C | Int J Infect Dis | 1 (China) | Retrospective | Case series | | 55 | 31 | 24 | Hospital | 55 (47.5–56.3) | 0 | 0 | - | - | 13 | 8 | National Health Commission of China |
| 3/7 | Zheng S | BMJ | 1 (China) | Retrospective | Cohort | | 96 | 38 | 58 | Hospital | 55 (47.5–56.3) | - | - | - | - | 25 | 49 | National Health Commission of China |
| 4/7 | Zhou F | Lancet | 2 (China) | Retrospective | Cohort | | 191 | 72 | 119 | Hospital | 56 (46–67) | 16 | 38 | - | - | - | - | National Health Commission of China |

NA: not applicable

**Table 2. Risk of bias assessment according the Newcastle Ottawa Score.**

| Author | Journal | Design | | Selection | | Comparability | Outcome | | | Score |
|---|---|---|---|---|---|---|---|---|---|---|
| | | | | 1 | 2 | 1 | 1 | 2 | 3 | |
| Chen G | *J Clin Invest.* | Retrospective | Case series | * | * | | * | | | 3/7 |
| Chen R | *Chest* | Retrospective | Cohort | * | * | | * | * | * | 5/7 |
| Chen T | *BMJ* | Retrospective | Case series | * | * | | * | * | | 4/7 |
| Chen X | *Clin Infect Dis.* | Retrospective | Case series | * | * | | * | | | 3/7 |
| Conversano A | *Hypertension* | Retrospective | Consecutive case series | | * | | * | * | | 3/7 |
| Gautret P | *Travel Med Infect Dis* | Retrospective | Cohort | * | * | | * | | | 3/7 |
| Guan W | *N Engl J Med* | Retrospective | Cohort | * | * | | * | | * | 4/7 |
| Hong KS | *Yonsei Med J* | Retrospective | Cohort | * | * | | * | | | 3/7 |
| Hou H | *Clin Exp Immunol* | Retrospective | Cohort | * | * | | * | | | 3/7 |
| Huang C | *Lancet* | Prospective | Obsevational cohort study | * | * | | * | | | 3/7 |
| Itelman E | *Isr Med Assoc J* | Retrospective | Cohort | * | * | | * | | | 3/7 |
| Korea—CDC | *Onsong Public Health Res Perspect* | Retrospective | Case series | | | | * | | | 1/7 |
| Lagi F | *Euro Surveill* | Retrospective | Cohort | * | | | * | | | 2/7 |
| Liu F | *Theranostics* | Retrospective | Cohort | * | * | | * | * | * | 5/7 |
| Liu F | *J Clin Virol* | Retrospective | Cohort | * | * | | * | | | 3/7 |
| Liu J | *EBioMedicine* | Retrospective | Cohort | * | * | | * | | * | 4/7 |
| Liu K-C | *Eur J Radiol* | Retrospective | Case series | * | * | | * | | | 3/7 |
| Liu Z | *Korean J Radiol* | Retrospective | Case series | * | * | | * | | | 3/7 |
| Marullo AG | *Minerva Cardioangiol* | Retrospective | Case series | | | | * | | | 1/7 |
| Pan L | *Am J Gastroenterol* | Retrospective | Case series | | | | * | * | * | 3/7 |
| Petrilli C | *BMJ* | Prospective | Obsevational cohort study | | * | | * | | | 2/7 |
| Shi H | *Lancet Infect Dis* | Retrospective | Case series | * | | | * | | | 2/7 |
| Sun B | *Emerg Microbes Infect* | Retrospective | Case series | * | | | * | | | 2/7 |
| Sun H | *J Am Geriatr Soc* | Retrospective | Case-control study | | | | * | * | * | 3/7 |
| Sun S | *Clin Chim Acta* | Retrospective | Case series | * | * | | * | | | 3/7 |
| Tang N | *J Thromb Haemost* | Retrospective | Cohort | | * | | * | * | * | 4/7 |
| Tian S | *J Infect* | Retrospective | Cohort | * | * | | * | | | 3/7 |
| To K | *Lancet Infect Dis* | Prospective | Obsevational cohort study | * | * | | * | | | 3/7 |
| Wan S | *Br J Haematol* | Retrospective | Case series | * | * | | * | * | * | 5/7 |
| Wang D | *Crit Care* | Retrospective | Case series | * | * | | * | * | * | 5/7 |
| Wang F | *Endocr Pract* | Retrospective | Case series | | * | | * | | | 2/7 |
| Wang J | *Clin Radiol* | Retrospective | Case series | * | * | | * | | | 3/7 |
| Wang L | *J Infect* | Retrospective | Consecutive case series | | * | | * | * | | 3/7 |
| Wang L | *Am J Nephrol* | Retrospective | Case series | * | * | | * | | | 3/7 |
| Wang M | *Aging (Albany, NY)* | Retrospective | Cohort | * | | | * | | * | 3/7 |
| Wu C | *JAMA Intern Med* | Retrospective | Cohort | * | * | | * | | | 3/7 |
| Xu Y-H | *J Infect* | Retrospective | Case series | * | * | | * | | | 3/7 |
| Yan Y | *BMJ Open Diabetes Res Care* | Retrospective | Case series | | * | | * | * | * | 4/7 |
| Yang A-P | *Int Immunopharmacol* | Retrospective | Case series | * | * | | * | | | 3/7 |
| Yang X | *Lancet Respir Med* | Retrospective | Cohort | | * | | * | * | * | 4/7 |
| Yang Y | *J Allergy Clin Immunol* | Retrospective | Case-control series | * | * | | * | | | 3/7 |
| Yuan M | *PloS One* | Retrospective | Consecutive case series | * | | | * | * | * | 4/7 |
| Zeng F | *J Med Virol* | Retrospective | Case series | * | | | * | | | 2/7 |
| Zhang G | *Respir Res* | Retrospective | Cohort | * | * | | * | | | 3/7 |
| Zheng C | *Int J Infect Dis* | Retrospective | Case series | * | * | | * | | | 3/7 |
| Zheng S | *BMJ* | Retrospective | Cohort | * | * | | * | | | 3/7 |
| Zhou F | *Lancet* | Retrospective | Cohort | * | * | | * | | * | 4/7 |

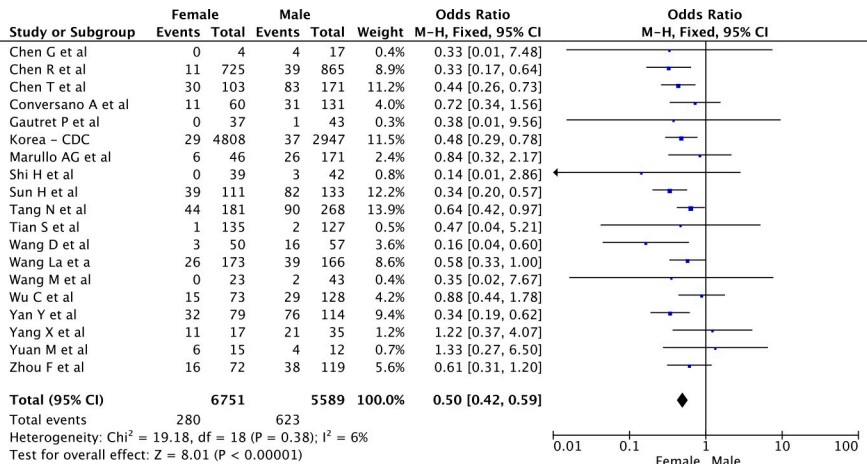

**Fig 2. Crude mortality risk according to the gender of COVID-19 patients.**

of severity, of the 23 studies, 14 used the definition of the National Health Commission of China [17, 19, 20, 24, 27, 29, 33, 36, 41, 42, 55, 56, 58, 62], 2 used the WHO interim guidelines [49, 52], 2 used the American Thoracic Society guidelines [35, 57], and the last 3 used a predetermined clinical presentation [44, 60, 63]. All these definitions and scores were based on clinical, biological and radiological parameters assessing respiratory failure, hypoxemia and disease progression. Only one study did not provide the exact definition of severe cases [39].

**RoB in the studies that contributed data to the analysis of severe disease presentation in men and women.** The assessment of the studies contributing to the severity analysis showed variability ranking from 2/7 to 5/7, as reported in Table 2.

**Severe disease presentation in men and women.** Overall, 355 (21%) women developed severe disease out of 1,721 women versus 500 (25%) men out of 1,970 men (OR 0.75 [0.60–0.93] for women versus men, p<0.001, see Fig 3). There was great variability in the rates of severe disease in men and women, with OR and 95% CI ranging from 0.23 [0.02, 2.73] to 2.14 [0.47, 9.74]. Heterogeneity reported as $I^2$ was reaching 28% for this analysis. The funnel plot (S1b Fig in S1 File) and the Egger test (p = 0.18) both indicated that there was no evidence for publication bias.

## Unadjusted rates of ICU admission

**Characteristics of the studies that contributed data to the analysis of mortality rates in men and women.** The ICU admission rates were reported separately for 4,222 men and

**Table 3. Certainty in overall mortality effect estimates using Grading of Recommendations Assessment, Development and Evaluation (GRADE) methods.**

| Certainty assessment | | | | | | Effect | Certainty | Importance |
|---|---|---|---|---|---|---|---|---|
| No. of studies | Study design | Risk of bias | Inconsistency | Indirectness | Imprecision | | | |
| 47 | Observational studies | Serious | Serious | Very Serious | Very Serious | OR 0.51 [0.42–0.61] | VERY LOW | CRITICAL |

CI, confidence interval; OR, odds ratio.

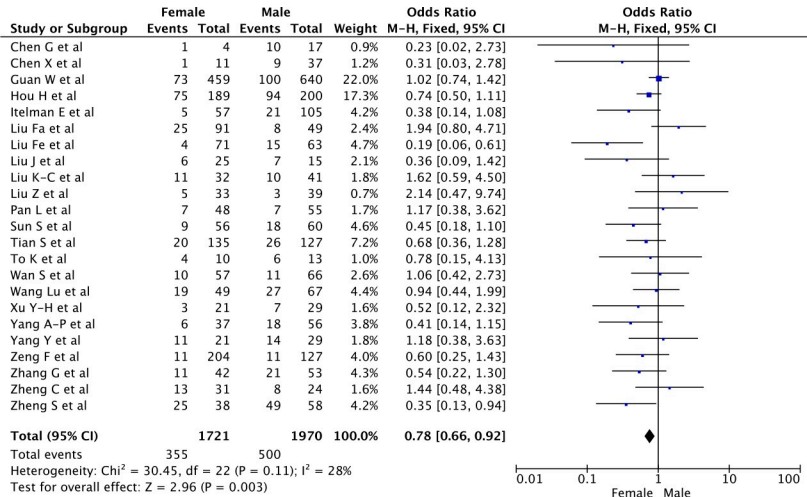

**Fig 3. Risk of severe presentation of the COVID-19 according to gender.**

3,859 women in 18 studies. Among the included studies 4 were prospective and observational, 5 were retrospective database studies, and 10 were case series. None of the studies aimed a-priori to compare differences between the sexes as the main outcome measure. Most of the studies included less than 100 patients. The characteristics and the main findings of the included studies are provided in Table 1.

**RoB in the studies that contributed data to the analysis of ICU admission rates in men and women.** The results of the quality assessment were highly variable ranging from 1/7 to 4/7 on the Newcastle Ottawa Scale (Table 2). The lowest scores are explained by the large number of case series included in this analysis.

**Unadjusted rates of ICU admission of men and women.** The proportion of men and women admitted to ICU was of 454 (12%) women and 931 (22%) men, (OR 0.45 [0.40–0.52], for women versus men, p<0.001, see Fig 4). Heterogeneity reported as $I^2$ was as low as 0% for this analysis. The funnel plot (S1c Fig in S1 File) and the Egger test (p = 0.51) both indicated that there was no evidence for publication bias.

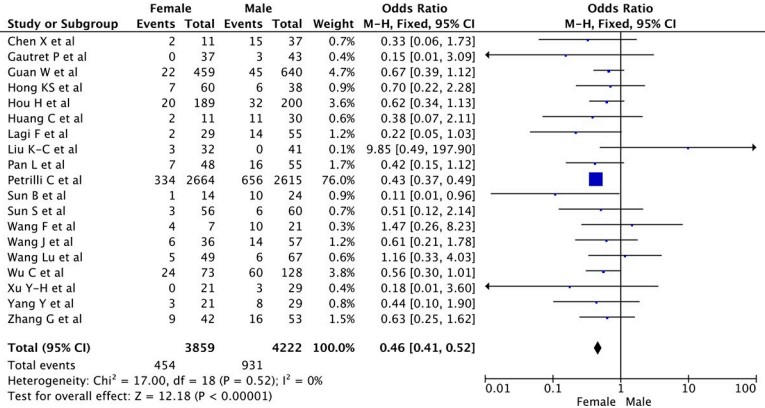

**Fig 4. Risk of ICU admission in COVID-19 patients according to gender.**

### IPDMA and adjusted outcomes

Despite repeated appeals, of all the authors contacted, only two responded to our request for data (the data was anonymized in both cases). This precluded performance of an IPDMA of adjusted outcomes; there were insufficient data on relevant variables (co-morbidities, treatment and age by vital status for males and females).

## Discussion

Unadjusted data suggest that men may have a higher risk of developing severe COVID-19 and also have higher associated death rates than women. Lacking detailed patient level data we were unable to determine whether this finding remains consistent after adjusting for the age and the burden of co-morbidity.

Several authors have suggested that men are more likely to be infected with COVID-19, especially after 50 years of age [64]. Alsan et al. reported that the incidence of COVID-19 in men was almost four times higher than in women (4.4% vs 1.2%) [65]. Here we report a similar total number of COVID-19 cases in men and women, suggesting that our findings are not directly related to a frequency argument.

Male-female differences in infectious diseases have already been reported and explored in the literature [9–11]. These differences are usually attributed to three determinants: differences in immune function associated with the X chromosome, the effects of sex hormones and gender-related behavior [12].

In the context of viral infections, men and women exhibit different immune reactions depending on the type of virus [66]. When infected with SARS-Cov, women usually have heightened immunity protecting them from severe forms of the disease, likely due to activation of the X regulatory genes, resulting in lower viral loads and higher CD4 T-cell counts [6, 67]. Women have additional immune characteristics which provide an advantage when exposed to viral infection compared to men. For example women have higher expression of Toll-like receptor 7 (TLR7) which is known to recognize viral RNA. They also produce more interferon-α, which is associated with the protection of lung tissue in animal models. Differences in production of IL-6 have also been observed between men and women [10].

Estradiol is believed to stimulate humoral and cell-mediated immune responses [68] and increase antibody production (73). Estradiol is also involved in regulating the expression of angiotensin converting enzyme 2 (ACE2), which is known for its protective role in acute respiratory distress syndrome [69, 70]. To date, there is no data available on gender-specific ACE2 expression in the lungs [71]. On the other side of the hormonal spectrum, testosterone is known to be immunosuppressive and decreased testosterone production is associated with elevated levels of pro-inflammatory cytokines [72]. Thus testosterone-related inhibition of the inflammatory response may dampen the antiviral response whereas an estrogen enhanced the increase in antibody and CD8 titers is likely to enhance the antiviral immune response.

Gender differences also exist in the dimension of social behaviour. Alsan et al. reported that men washed their hands 3.8 times per day less than women in a national US survey. Men were also more inclined to go outside than women during lockdown, which may have increased their exposure to SARS-CoV-2 [65]. Comorbidities such as obesity, diabetes and hypertension, which are unequally distributed between men and women at different ages, may also affect the course of disease [73, 74].

Finally, the role of the virus itself should not be underestimated; for example, influenza has higher morbidity and mortality in women than in men [8].

Despite our exhaustive data collection, our analyses are probably tinged with significant selection bias. The studies included report data mainly from China; i.e. the data presented are

from the early stage of the outbreak. Such bias could explain the high mortality rates found in our study. Nonetheless, more recent data from Italy also showed higher mortality rates among men [75]. Men also had a higher rate of ICU admission in the US than women (OR = 2.0; 95% CI: 1.3–3.2) [76]. However, while this finding may indicate worse disease severity in men, it could also stem from admission bias. Another limitation of our study is that most of the studies included were retrospective and had a short follow-up period which may lead to underestimation of long-term mortality. The mortality reported in the included studies was mostly in-hospital mortality (except for one study) and ICU mortality was specifically reported in only five studies. These considerations are reflected in the RoB assessment where they negatively affected study ratings, leading to a very low certainty of evidence. But the greatest limitation of our study is that these data could not be adjusted for age, co-morbidities or treatment.

To conclude, COVID-19 may be associated with greater disease severity and higher mortality rates among males. However, this preliminary finding remains unproven until such time that adjusted analyses can be conducted. Our results are in line with previous studies on viruses and are also supported by a strong biological rationale, but the level of evidence is limited by the poor quality of the available data. We urge researchers to conduct further studies and request sharing of data on gender dimorphism in COVID-19 in order to provide more robust evidence on this topic.

## Supporting information

**S1 Checklist. PRISMA 2009 checklist.**
(DOC)

**S1 File.**
(DOCX)

## Author Contributions

**Conceptualization:** Ines Lakbar, Sharon Einav, Marc Leone.

**Data curation:** Ines Lakbar, David Luque-Paz.

**Formal analysis:** Ines Lakbar, David Luque-Paz.

**Software:** Ines Lakbar.

**Supervision:** Sharon Einav, Marc Leone.

**Writing – original draft:** Ines Lakbar, David Luque-Paz.

**Writing – review & editing:** Jean-Louis Mege, Sharon Einav, Marc Leone.

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
