## [Decision Letter · Decision Letter 0]

13 Oct 2020

PONE-D-20-25279

COVID-19 gender susceptibility and outcomes: a systematic review

PLOS ONE

Dear Dr. Lakbar,

Thank you for submitting your manuscript to PLOS ONE. After careful consideration, we feel that it has merit but does not fully meet PLOS ONE’s publication criteria as it currently stands. Therefore, we invite you to submit a revised version of the manuscript that addresses the points raised during the review process.

ACADEMIC EDITOR:

Two reviewers, experts in the field, agreed on the need of underlying the low quality of the available data. I also ask the authors to remark further that no adjusted analysis has been performed for the quantitative synthesis and this further limit the possibility to make reliable conclusions.

We look forward to receiving your revised manuscript.

Kind regards,

Andrea Cortegiani, M.D.

Academic Editor

PLOS ONE

Journal Requirements:

2.Thank you for stating the following in the Competing Interests section:

[IL, DLP, SE and JLM have no conflict of interest to disclose.

ML served as speaker for MSD, Pfizer and as consultant for Amomed, Aguettant and Gilead].

Additional Editor Comments (if provided):

Reviewers' comments:

Reviewer's Responses to Questions

**Comments to the Author**

1. Is the manuscript technically sound, and do the data support the conclusions?

Reviewer #1: Yes

Reviewer #2: Yes

Reviewer #3: Partly

2. Has the statistical analysis been performed appropriately and rigorously? 

Reviewer #1: Yes

Reviewer #2: Yes

Reviewer #3: Yes

3. Have the authors made all data underlying the findings in their manuscript fully available?

Reviewer #1: Yes

Reviewer #2: Yes

Reviewer #3: Yes

4. Is the manuscript presented in an intelligible fashion and written in standard English?

Reviewer #1: Yes

Reviewer #2: Yes

Reviewer #3: Yes

5. Review Comments to the Author

Reviewer #1: I have had the opportunity to review the following manuscript: COVID-19 gender susceptibility and outcomes: a systematic review. There has been a significant amount of publications regarding the gender susceptibility and outcomes in these patients and this systematic review is hugely needed.

Lakbar and colleagues evaluated mainly studies from China and this is the main limitation of the paper as this might limit that is being applied to other regions of the world. I however understand that the authors had to limit the search of manuscripts to a certain time point and more papers from western countries have been published

In the PICO question O is obviously outcomes but I recommend the authors to be more specific about "the outcome". Mortality, ICU admission, etc?

In results there is data that should be explained such as "consisting of 11,176 women and 10,278 men (17,18,27–36,19,37–46,20,47– 148 56,21,57–61,22–26)" please explain further what's in the brackets for the non expert reader

In mortality rates, the data are fair enough but could you report ICU mortality? As this represents those patients acutely ill it would benefit data interpretation

Discussion is fine and well balanced.

Reviewer #2: Thank you for inviting me to review this meta-analysis

In this manuscript, the authors investigated the different outcomes across genders among patients diagnosed with CONVID-19. The authors observed higher unadjusted mortality among men. No adjustment could be made due to a lack of available data.

The manuscript is clear, the different hypotheses discussed, and the limits of the investigation well reported.

Major comments:

The significant limits are inherent to the quality of available data, which is overall very poor, limiting the results' impact. Impact of underlying comorbidities, age, the timing of first medical contact, treatments, etc could not be taken into account, much limiting the significance of the investigation.

- In the methods, the authors report that authors of original studies were contacted for individual patient metanalyses, but there is no further mention of this analysis.

databases up or observational studies

-I strongly suggest to edit the text due to numerous grammatical mistakes throughout the manuscript,

Reviewer #3: Thank you for the opportunity to review this manuscript. In this paper, the authors conduct a systematic review of existing published data to evaluate the impact of gender on outcomes for patients infected with the severe acute respiratory syndrome coronavirus-2 (SARS-CoV-2). The selected literature is reasonable and the conclusions are mostly sound. I do have some specific comments that I hope may improve the quality of the manuscript and its conclusions.

Abstract:

Line 26 – recommend rephrasing the line as “sepsis, influenza, and severe coronavirus infections including SARS-CoV and MERS-CoV”.

Line 30 – methods section. What was the total number of patients included in the analysis?

Line 33 – results section. The OR of 0.51 is presented ambiguously; from the context, I assume that the authors mean that the OR of mortality for women is 0.51 compared with men, but phrasing does not make this clear. I would make similar comments about the OR for severe disease and ICU admission in men vs. women.

Line 38 – conclusions. I would say “worse outcomes” rather than “poorer outcomes”. Also, “data” is plural in English, so the second sentence should say “until more detailed data are provided…”.

Introduction:

Lines 33-34: “Gender dimorphism” as a term usually refers to differing physical characteristics between the sexes, e.g., genitalia, body mass, etc. Its use in this context is non-standard. I recommend an alternate choice of terms, such as “differing severity and outcomes between the genders” or something similar.

Line 34: Disease incidence, rather than prevalence, is a more-suitable term when evaluating rates of acute infectious diseases without a chronic component.

Line 46: Suggest “highly-pathogenic coronaviruses” to distinguish between typical seasonal coronaviruses and COVID/SARS/MERS.

Line 47, line 48: “SARS-CoV-1” is not proper terminology. Use “SARS-CoV” without the digit.

Line 49: Use “worse” instead of “poorer”.

Methods:

Line 70: Say “Methods” at the section title, not “method”.

Line 91: Repeated use of “and/or” is awkward. I recommend “confirmed disease, admitted to the hospital, and/or admitted to the intensive care unit (ICU), as well as their mortality rates.”

The remainder of the results section is sound, and I have no comments on a well-planned study design.

Results:

Line 149: There is no hyphen in “Hong Kong”.

Line 192: The authors write “355 (21%) women developed severe disease versus 500 (25%) men”. What are the denominators here? 21% of what? Of all cases in women and men? And what is the definition of severe disease being used in these cohorts? Is it the standard WHO definition (pneumonia with hypoxemia) or is it more associated with critical illness and ICU admission?

Discussion:

Line 242: The authors state that women with SARS-CoV infection “usually have heightened immunity”. This term lacks precision; “heightened immunity” could refer to decreased susceptibility to severe disease, or it could indicate increased inflammatory levels leading to a dysregulated immune response and worse outcomes. 

6. PLOS authors have the option to publish the peer review history of their article (what does this mean?). If published, this will include your full peer review and any attached files.

Reviewer #1: No

Reviewer #2: No

Reviewer #3: Yes

---

## [Author Response · Author response to Decision Letter 0]

20 Oct 2020

PLOS ONE submission PONE-D-20-25279

Toulouse, October the 13th, 2020

Object: Response to Reviewers

Dear Editor, dear Reviewers,

We thank you for your comments and your questions. Please find our response below. 

Academic editor: Two reviewers, experts in the field, agreed on the need of underlying the low quality of the available data. I also ask the authors to remark further that no adjusted analysis has been performed for the quantitative synthesis and this further limit the possibility to make reliable conclusions.

Dear academic editor, we thank you for your time and attention, and we are grateful that you give us the opportunity to correct and improve our manuscript.

 These two points correspond to the two major conclusions of our paper: the data are poor and the analyses unadjusted. Our paper is a call for the production of detailed and robust data.

We hope that the revised version will reach your expectations. 

Sincerely yours, 

Inès Lakbar

Reviewer #1: I have had the opportunity to review the following manuscript: COVID-19 gender susceptibility and outcomes: a systematic review. There has been a significant amount of publications regarding the gender susceptibility and outcomes in these patients and this systematic review is hugely needed. 

Dear Reviewer 1,

That is our pleasure to respond to each of your comment. We thank you for this relevant review that made us possible to improve our manuscript.

Sincerely yours,

Inès Lakbar

Lakbar and colleagues evaluated mainly studies from China and this is the main limitation of the paper as this might limit that is being applied to other regions of the world. I however understand that the authors had to limit the search of manuscripts to a certain time point and more papers from western countries have been published

The last research was conducted on June 1st and at that time most of the included studies originated from different hospitals in China (n=38), one originated from Hong Kong, two from South Korea, five from Europe and North America and one from Israel. The sensitivity analysis showed no difference in the main outcomes of mortality, ICU admission and severity of the disease when non-Chinese studies were excluded (results are reported in the supplementary material). We have addressed this issue as a selection bias in our discussion section, and we call for the production of detailed and robust data.

In the PICO question O is obviously outcomes but I recommend the authors to be more specific about "the outcome". Mortality, ICU admission, etc?

A sentence has been added and specifies the outcomes: “PICO question: We sought to study whether among adult patients with COVID-19 (P) women (“I”) differ from men (C) with regards to disease characteristics and outcomes (mortality, severity and ICU admission rates) (O).” Therefore, outcomes here are mortality rates, severity of the disease and ICU admission rates.

In results there is data that should be explained such as "consisting of 11,176 women and 10,278 men (17,18,27–36,19,37–46,20,47– 148 56,21,57–61,22–26)" please explain further what's in the brackets for the non-expert reader

The numbers in the brackets correspond to the scientific articles included and refer to the bibliography. We apologize for this typographical mistake. A correction has been made to make it easier to read: “A total of 2,379 studies were screened, of which 47 contributed data to our analysis for a total of 21,454 patients consisting of 11,176 women and 10,278 men (17–61).”

In mortality rates, the data are fair enough but could you report ICU mortality? As this represents those patients acutely ill it would benefit data interpretation

Mortality rates in ICU were reported in only five studies, two of which only reported data for ICU patients. This represents a total of 577 patients and 188 deaths. As compared to crude mortality rates, reported separately for 5,589 men and 6,751 women in 19 studies and ICU admission rates reported separately for 4,222 men and 3,859 women in 18 studies, these data seemed to be too insufficient to be reported in the manuscript. 

A sentence has been added in the limitation section to underline this point, which you have very rightly pointed out: ”The mortality reported in the included studies was mostly in-hospital mortality (except for one study) and ICU mortality was specifically reported in only five studies.”

Discussion is fine and well balanced.

We thank you once again for your reviewing, we hope we have answered all your queries and comments. 

Reviewer #2: Thank you for inviting me to review this meta-analysis. In this manuscript, the authors investigated the different outcomes across genders among patients diagnosed with CONVID-19. The authors observed higher unadjusted mortality among men. No adjustment could be made due to a lack of available data. The manuscript is clear, the different hypotheses discussed, and the limits of the investigation well reported.

Dear Reviewer 2,

Your comments helped us to improve the quality of our manuscript. We hope that the revised version will reach your expectations. 

Sincerely yours,

Inès Lakbar

Major comments:

The significant limits are inherent to the quality of available data, which is overall very poor, limiting the results' impact. Impact of underlying comorbidities, age, the timing of first medical contact, treatments, etc could not be taken into account, much limiting the significance of the investigation.

- In the methods, the authors report that authors of original studies were contacted for individual patient metanalyses, but there is no further mention of this analysis.

Line 221, a paragraph explains that the individual patient data metanalysis could not be performed as only two authors responded to our request for data.

“Despite repeated appeals, of all the authors contacted, only two responded to our request for data (the data was anonymized in both cases). This precluded performance of an individual patient data metanalysis of adjusted outcomes; there was insufficient data on relevant variables (co-morbidities, treatment and age by vital status for males and females).”

databases up or observational studies

-I strongly suggest to edit the text due to numerous grammatical mistakes throughout the manuscript,

As you suggested, we performed a professional English editing. 

Thank you very much for your time and attention to our manuscript. 

Reviewer #3: Thank you for the opportunity to review this manuscript. In this paper, the authors conduct a systematic review of existing published data to evaluate the impact of gender on outcomes for patients infected with the severe acute respiratory syndrome coronavirus-2 (SARS-CoV-2). The selected literature is reasonable and the conclusions are mostly sound. I do have some specific comments that I hope may improve the quality of the manuscript and its conclusions.

Dear Reviewer 3,

We thank you for your attention to this paper, and we have scrupulously followed your advice and corrections. We hope that these improvements will bring this manuscript to the level of your expectations.

Sincerely yours,

Inès Lakbar

Abstract:

Line 26 – recommend rephrasing the line as “sepsis, influenza, and severe coronavirus infections including SARS-CoV and MERS-CoV”.

Thank you for your suggestion, the sentence has been rephrased this way “sepsis, influenza, and severe coronavirus infections including SARS-CoV and MERS-CoV”.

Line 30 – methods section. What was the total number of patients included in the analysis?

The number of patients has been added as suggested. 

Line 33 – results section. The OR of 0.51 is presented ambiguously; from the context, I assume that the authors mean that the OR of mortality for women is 0.51 compared with men, but phrasing does not make this clear. I would make similar comments about the OR for severe disease and ICU admission in men vs. women.

The sentence has been rephrased “The unadjusted mortality rates of men were higher than those of women, with a mortality OR 0.51 [0.42, 0.61] (p<0.001) for women.”

Line 38 – conclusions. I would say “worse outcomes” rather than “poorer outcomes”. Also, “data” is plural in English, so the second sentence should say “until more detailed data are provided…”.

The conclusion has been rephrased as suggested. Thank you for your advice.

Introduction:

Lines 33-34: “Gender dimorphism” as a term usually refers to differing physical characteristics between the sexes, e.g., genitalia, body mass, etc. Its use in this context is non-standard. I recommend an alternate choice of terms, such as “differing severity and outcomes between the genders” or something similar.

The word dimorphism has been replaced by the word differences. Thank you for this correction.

Line 34: Disease incidence, rather than prevalence, is a more-suitable term when evaluating rates of acute infectious diseases without a chronic component.

Prevalence has been replaced by incidence. Thank you for this suggestion. 

Line 46: Suggest “highly-pathogenic coronaviruses” to distinguish between typical seasonal coronaviruses and COVID/SARS/MERS.

The word “highly” has been added as suggested

Line 47, line 48: “SARS-CoV-1” is not proper terminology. Use “SARS-CoV” without the digit.

The correction has been made.

Line 49: Use “worse” instead of “poorer”.

Poorer has been replaced by worse. Thank you once again.

Methods:

Line 70: Say “Methods” at the section title, not “method”.

The correction has been done, thank you. 

Line 91: Repeated use of “and/or” is awkward. I recommend “confirmed disease, admitted to the hospital, and/or admitted to the intensive care unit (ICU), as well as their mortality rates.”

The sentence has been rephrased as suggested.

The remainder of the results section is sound, and I have no comments on a well-planned study design.

Results:

Line 149: There is no hyphen in “Hong Kong”.

The correction has been done.

Line 192: The authors write “355 (21%) women developed severe disease versus 500 (25%) men”. What are the denominators here? 21% of what? Of all cases in women and men? And what is the definition of severe disease being used in these cohorts? Is it the standard WHO definition (pneumonia with hypoxemia) or is it more associated with critical illness and ICU admission?

Denominators, mentioned in the descriptive paragraph (5 lines above) have been added again line 192. 

For the definition of severity, this paragraph has been added: “As for the definition of severity, of the 23 studies, 14 used the definition of the National Health Commission of China (17,19,55,56,58,62,20,24,27,29,33,36,41,42), 2 used the WHO interim guidelines (49,52), 2 used the American Thoracic Society guidelines (35,57), and the last 3 used a predetermined clinical presentation (44,60,63). All these definitions and scores were based on clinical, biological and radiological parameters assessing respiratory failure, hypoxemia and disease progression. Only one study did not provide the exact definition of severe cases (39).”

Thank you for giving us the opportunity to clarify the main message here.

Discussion:

Line 242: The authors state that women with SARS-CoV infection “usually have heightened immunity”. This term lacks precision; “heightened immunity” could refer to decreased susceptibility to severe disease, or it could indicate increased inflammatory levels leading to a dysregulated immune response and worse outcomes. 

The sentence has been modified as follow: “When infected with SARS-Cov, women usually have heightened immunity protecting them from severe forms of the disease”

Thank you for your attention with regards to our manuscript, 

Sincerely yours,

Inès Lakbar

---

## [Editor Report · Decision Letter 1]

22 Oct 2020

COVID-19 gender susceptibility and outcomes: a systematic review

PONE-D-20-25279R1

Dear Dr. Lakbar,

We’re pleased to inform you that your manuscript has been judged scientifically suitable for publication and will be formally accepted for publication once it meets all outstanding technical requirements.

Kind regards,

Andrea Cortegiani, M.D.

Academic Editor

PLOS ONE
---

## [Editor Report · Acceptance letter]

26 Oct 2020

PONE-D-20-25279R1 

COVID-19 gender susceptibility and outcomes: a systematic review 

Dear Dr. Lakbar:

I'm pleased to inform you that your manuscript has been deemed suitable for publication in PLOS ONE. Congratulations! Your manuscript is now with our production department. 

Kind regards, 

on behalf of

Dr. Andrea Cortegiani 

Academic Editor

PLOS ONE